# Co-Design Practices in Diet and Nutrition Research: An Integrative Review

**DOI:** 10.3390/nu13103593

**Published:** 2021-10-14

**Authors:** Brenda S. J. Tay, David N. Cox, Grant D. Brinkworth, Aaron Davis, Sarah M. Edney, Ian Gwilt, Jillian C. Ryan

**Affiliations:** 1Nutrition & Dietetics, College of Nursing & Health Sciences, Flinders University, GPO Box 2100, Adelaide, SA 5001, Australia; 2Health and Biosecurity, Commonwealth Scientific and Industrial Research Organisation, 13 Kintore Avenue, Adelaide, SA 5000, Australia; david.cox@csiro.au (D.N.C.); grant.brinkworth@csiro.au (G.D.B.); jillian.ryan@csiro.au (J.C.R.); 3UniSA Creative, University of South Australia, Adelaide, SA 5001, Australia; aaron.davis@unisa.edu.au (A.D.); ian.gwilt@unisa.edu.au (I.G.); 4Physical Activity and Nutrition Determinants in Asia (PANDA) Programme, Saw Swee Hock School of Public Health, National University of Singapore, Singapore 119077, Singapore; sarah.edney@nus.edu.sg

**Keywords:** co-design, participatory research, nutrition, diet, intervention

## Abstract

Co-design, the method of involving users, stakeholders, and practitioners in the process of design, may assist to improve the translation of health evidence into tangible and acceptable intervention prototypes. The primary objective of this review was to identify and describe co-design techniques used in nutrition research. The secondary objective was to identify associations between co-design techniques and intervention effectiveness. An integrative review was performed using the databases Emcare, MEDLINE, PsycINFO and Google Scholar. Eligible studies included those that: (1) utilised participatory research or co-design techniques, (2) described development and/or evaluation of interventions aimed at improving dietary behaviours or nutrition, and (3) targeted community-dwelling adults aged ≥18 years. We identified 2587 studies in the initial search and included 22 eligible studies. There were 15 studies that utilised co-design techniques, with a strong focus on engagement of multiple stakeholder types and use of participatory research techniques. No study implemented a complete co-design process. Most studies (14/15) reporting outcomes reported positive health (maximum *p* < 0.001) or health behaviour outcomes attributed to the intervention; hence, associations between co-design techniques and effectiveness could not be determined. Currently published intervention studies have used participatory research approaches rather than co-design methods. Future research is required to explore the effectiveness of co-design nutrition interventions.

## 1. Introduction

Over the past half-century, dietary intakes have changed dramatically, with increased consumption of processed foods containing added sodium, unhealthy fats, and refined carbohydrates/sugars [1]. Men and women across all age groups consume high amounts of discretionary (unhealthy) food with underconsumption of fruits and vegetables relative to health guidelines. These dietary factors are major drivers for common chronic conditions including cancer, heart disease and Type 2 diabetes [2], which are leading contributors to early death, illness, and disability [3]. Improving dietary behaviour is a cornerstone in the prevention and treatment of chronic diseases [4], but remains a significant challenge [4]. To effectively achieve dietary behaviour change, interventions must be embedded in best practice, associated with effectiveness, and be relevant and appealing to target populations to facilitate successful translation into practice. Hence, nutrition interventions are increasingly focused on patient-centred models [4].

Person- or community-centred care is the foundation of dietetic practice [5]. This refers to healthcare providers building relationships with people and their communities to manage health conditions in a personalised approach that provides equal sharing of power [6]. In this manner, public health (nutrition) research interventions should consider a similar approach, including “bottom-up” participatory research or participatory action research (PAR) designs. The benefits of PAR are widely acknowledged and include the development of research outputs closely aligned to community needs, while helping to build community capacity and promoting research equity [7]. Notably, PAR defies traditional “top-down” research methods to disassemble traditional power imbalances between participants and researchers.

Co-design, also known as co-creation, co-production, or participatory design, in a healthcare setting, refers to the integration of design thinking, stakeholder experiences, scientific evidence and participatory principles in the collaborative design of local solutions to local problems [8,9,10]. Co-design is considered to produce solutions based on understanding of the local context to meet the needs of all stakeholders [11], offers insights into the lived experience of the public and helps to answer the why questions as opposed to science-based research, which predominantly looks at what is happening. Therefore, co-design may have greater acceptance by providers and target users [9], and offer a more sustainable and effective translation approach into clinical practice. Furthermore, controlled trials provide rigorous evidence of the inherent value of community inclusion in public health research processes, particularly for increasing the effectiveness of interventions, achieving local customisation and strong community engagement [12], and improving the quality and appropriateness of study design [13].

Co-design research methods can also be effective in helping to overcome barriers to translation and improving the uptake and effectiveness of nutrition interventions, as it is unclear to what extent co-design has been incorporated into nutrition research, nor what the benefits are. However, studies evaluating the effectiveness of co-design appear to be scarce [8,14], and the effectiveness of co-design methodological techniques used in different research disciplines remains unfixed. Co-design and PAR approaches have also been the subject of intense debate with key criticisms including poor reporting practices [15] and tokenism: “small-scale, poorly funded and with limited incentives” in co-design activities [16]. Furthermore, there remains limited published studies, systematically reviewing the participatory design of nutrition/diet-based interventions. Hence, the purpose of this study was to conduct an integrative, systematic review to identify and describe participatory and co-designed methodological techniques previously used in nutrition research and to identify any associations between the use of participatory or co-design techniques and intervention effectiveness. This will assist to guide future nutrition research that deploys co-design or participatory research methods. 

## 2. Methods

Ethics approval was not obtained for this study since human participants were not involved. An integrative review using a systematic review search approach was undertaken following Preferred Reporting Items for Systematic Reviews and Meta-Analyses (PRISMA) guidelines. A protocol for this review was written, agreed upon by all co-authors and registered with Open Science Framework (osf.io/s8cv7). All stages of literature searching and screening were conducted by the first author (B.S.J.T.) with assistance from the co-authors as specified.

An integrative review approach was used as it enables systematic and rigorous review of studies that contain diverse methodologies, for example, experimental, non-experimental, quantitative, and qualitative work [17]. Integrative reviews share common search strategies for promoting rigor associated with systematic reviews, but diverge at the point of data analysis, with integrative reviews drawing upon inductive techniques such as the identification of noting patterns and themes, seeing plausibility, clustering, counting, making contrasts and comparisons, discerning common and unusual patterns, subsuming particulars into general, noting relations between variability, finding intervening factors, and building a logical chain of evidence [17].

### 2.1. Eligibility Criteria

Eligibility criteria are outlined below, according to the PICOS structure. 

#### 2.1.1. Population

Target population was restricted to community-dwelling adults who were 18 years old or older. Studies including children and adolescents were excluded due to vast differences in characteristics, learning and behavioural issues compared with adults. Studies conducted in any geographical locations including metropolitan, rural, and remote areas and online settings were included. Gender and health status of the study population was not limited, i.e., healthy populations or individuals with a specific health condition were included. Study populations excluded were ex vivo, in vivo, and in vitro studies and studies where the participants were not humans, or animal models. 

#### 2.1.2. Intervention 

Eligible studies reported an intervention that aimed to improve dietary behaviours or any aspect of nutrition—for example, interventions that aim to increase vegetable consumption, examine the effects of a specific food, ingredient, or compound on a health-related outcome, or to encourage compliance with an entire diet. Mixed interventions (interventions targeting dietary and other health components risk factors such as physical activity) were also included. Interventions could be delivered via any format including digital, face-to-face, or mixed. To be included, co-design or participatory research methods must have been used to develop the intervention. The definition of co-design varies with different authors; however, the general rule is that the research methods should include active collaboration between participants, researchers and other relevant stakeholders in the process of intervention design [18]. To exclude interventions with limited participant involvement in their development, only studies classified as having involved “collegiate”, “collaborative”, or “consultative” participation were included. Classification was based on the seminal definitions of the four modes of participation described by Cornwall and Jewkes [19], built upon by Biggs [19,20]. Contractual participation is considered to reflect shallow participation, while consultative research approaches genuine participation, and collaborative and collegiate meet standards for genuine participation [19].

#### 2.1.3. Control or Comparator 

Studies were not limited based on control group or comparator.

#### 2.1.4. Outcomes 

Studies that described either the design, development or evaluation of a co-designed dietary behaviour intervention were eligible. Therefore, eligible outcomes include:

Description of characteristics of an intervention: Studies that describe the participatory, co-design, or stakeholder engagement techniques that they applied in the development of a diet or nutrition intervention were included. Hence, papers describing study protocols were eligible.

Outcomes: Studies were included if they reported dietary behavioural outcomes (e.g., increased fruit and vegetable intake) and/or health outcomes (e.g., weight loss). Study outcomes could also be the development of a co-designed intervention, a new tool, or specific intervention components. Qualitative studies undertaken to directly inform the design of a specific intervention were included, while qualitative or consultative research for general knowledge purposes was ineligible (e.g., identifying barriers and facilitators to dietary behaviours).

#### 2.1.5. Study Design 

As this is an integrative review, any type of primary study (qualitative or quantitative) was eligible. Studies of any sample size, protocols for planned studies and studies that reported descriptions of the co-design process or methods without outcome evaluations were included. Studies that reported process evaluations of the intervention were also eligible. Other publication types such as review articles, opinions or editorials and conference abstracts of less than 1000 words were excluded.

To be included, study methodologies had to meet collegiate, collaborative, or consultative levels of participation [18]. Studies that met just the contractual level were excluded (see Table 1).

### 2.2. Data Sources and Search Strategy

The search strategy centred upon two concepts: (1) co-design and (2) dietary intervention, and was developed with input from an academic librarian at Flinders University (Adelaide, South Australia). After conducting experimental searches, the proximity searching technique was used for the “dietary intervention” concept to improve search precision and reduce the number of results produced. In Google Scholar where proximity searching is not applicable, “dietary intervention” was collapsed into two distinct concepts, i.e., “diet” and “intervention”. Hence, the search strategy used in Google Scholar was the combination of synonyms of “co-design”, “participatory action research”, “diet” and “intervention”.

The systematic search was conducted on 11 August 2020 using four electronic databases (Emcare, MEDLINE, PsycINFO and Google Scholar). Consistent with evidence regarding optimal coverage for health and medical topics, Emcare, MEDLINE and PsycINFO databases were searched with the following filters applied: (1) humans, (2) full text available, (3) published date 2010–2020 and (4) English language [21]. Search terms included synonyms of the search concepts “co-design” and “dietary intervention” (see Table 2) and were searched in all fields.

### 2.3. Data Extraction and Synthesis

#### 2.3.1. Selection of Studies

All references from searches of electronic databases were imported into the systematic review software, Covidence, for screening. Two authors (B.S.J.T. and J.C.R.) screened 20% of the titles and abstracts in duplicate according to the agreed eligibility criteria. The first author (B.S.J.T.) screened the remaining 80% of articles independently. Full-text screening was undertaken in duplicate, independently, by the same individuals and any disagreements were resolved through discussion with an independent adjudicator.

#### 2.3.2. Classification of Studies Based on Modes of Participation

Due to variation in participation across different studies further screening was conducted to classify the extent to which participatory techniques were utilised in each study according to the four modes of participation described by Cornwall and Jewkes [19]. Since contractual research involves only minor and superficial consultation with participants, articles were only included in this review if the intervention design reached collegiate, collaborative, and consultative participatory standards. Two authors (B.S.J.T. and J.C.R.) classified the studies in independent duplicate and any conflicts were discussed and resolved.

#### 2.3.3. Data Extraction and Management

Data were extracted into a purpose-developed data extraction table. Information regarding the studies’ characteristics (aim, participants, inclusion of other stakeholders, setting, intervention, main outcome or finding, PAR standard reached) and co-design methods (theoretical framework, co-design approach, data collection/analysis techniques, research stage at which participant feedback was sought, and extent of engagement) were included. 

#### 2.3.4. Sufficiency of Reporting

An assessment of sufficiency of reporting was undertaken using an adapted version of the eight-item checklist for reporting non-pharmacological interventions, originally adapted from the Consolidated Standards of Reporting Trials (CONSORT) checklist [14,22,23]. Studies were scored against items relating to the (1) setting, (2) stakeholders, (3) facilitators, (4) co-design methods, (5) materials, (6) length of design and sessions, (7) interval and frequency of sessions and (8) description of the overall co-design process.

## 3. Results

Identification and selection of studies is summarised in Figure 1. After full-text screening, 36 studies were eligible. Following further screening to exclude contractual modes of participation, 22 studies were included in this review. 

### 3.1. Characteristics of Included Studies

Study characteristics are depicted in Table 3.

#### 3.1.1. Country

The 22 included studies were from ten countries: seven from the USA [28,33,34,41,45,47,49]; three each from Australia [24,25,27] and Sweden [40,44,48]; two from the Netherlands [29,30] and the UK (Bangladeshi migrant community) [36,51]; and one each from Canada [26], Brazil [39], Belgium [43], Iran [46] and Malaysia [50].

#### 3.1.2. Study Design

None of the studies specifically reported to have used co-design methods. However, they each used a participatory technique within the context of intervention development. Eleven studies reached a consultative (second lowest) level of participation [40,41,43,44,45,46,47,48,49,50,51], eight studies reached collaborative (second highest) levels [27,28,29,30,33,34,36,39], and three reached collegiate (highest) levels of participation [24,25,26].

Studies included qualitative (*n* = 8), quantitative (*n* = 11) and mixed methods research designs (*n* = 3). Studies using collegiate and collaborative approaches were mostly qualitative or mixed methods studies (8/8), whereas the consultative studies were mostly quantitative studies (10/15). 

#### 3.1.3. Participants

Participant demographics varied across studies. Participants included Indigenous adults from Australia and Canada [24,26], community-dwelling older adults [27,29,50], African American women and children [28,33,35], health professionals [25,29,50,51], workplace employees [32,39], individuals with chronic disease [25,29,36,44,45,47,48,50,51], healthy individuals [43,49,51], weight loss program clients [40,46], and women with diet-controlled gestational diabetes mellitus [41]. The age of participants ranged from 18 to 65 years or more. 

#### 3.1.4. Other Stakeholders Involved in the Intervention Development Process

Of the included studies, 13 reported involving other stakeholders other than end-users/members of the target population in the intervention development process. Stakeholders consisted of community organisations [26,36,51], health professionals [29,36,39,45,50,51], food retail [26], food product developers [29], catering providers [32,39,49], public enterprise [32], primary healthcare [26], health promotion experts [27], students from disciplines other than nutrition [27,28], tutors [44], visual communication experts [50] and staff from a software company [29,36,39,45,50,51].

#### 3.1.5. Theoretical Frameworks 

Twelve studies reported using a theory-based framework in developing the intervention. Social Cognitive Theory [52] was the most common theory used (*n* = 3), followed by PAR (*n* = 2) and Intervention Mapping (*n* = 2). 

#### 3.1.6. Recruitment Methods

Recruitment methods varied. Three studies [36,44,48] did not report methods of participant recruitment and only two studies [33,47] reported providing reimbursement to participants for their contribution.

#### 3.1.7. Participatory Design Methods and Processes

The 11 techniques used to consult stakeholders’ perspectives in the included studies are summarised in Table 3. The three collegiate studies, which represent the highest level of stakeholder engagement, utilised various methods including the photo-voice method [24], workshops [25,26], interviews [26], stakeholder meetings [26] and dietary assessment [26]. Across remaining studies, focus groups were most common (*n* = 9) [27,28,29,32,33,35,36,39,45], followed by tailoring of intervention content to individual’s preferences, characteristics, or needs (*n* = 5) [40,43,46,48,49], and surveys (*n* = 5) [29,33,41,47,50].

#### 3.1.8. Extent of Participation

The extent to which participants had input was classified according to the six intervention development phases identified by Eyles et al. [14]. All, except two studies, assessed user needs to inform intervention focus. Six studies had end-user input in pilot/real-world testing, whereas four studies included end-users in prototype testing. Four studies assessed background knowledge and evidence, two studies assessed user needs to inform technology, and two studies involved participants in developing intervention content.

#### 3.1.9. Data Analysis Methods

Thematic analysis was the most comment approach to data analysis [24,25,26,29,32,33,35]. Other analysis methods used were content analysis [28] and dietary analysis [26,39,45,47].

#### 3.1.10. Intervention Effectiveness

Fourteen out of 22 studies reported outcomes, with 13 studies reporting statistically significant changes in diet- or nutrition-related outcomes or behaviour attributed to the intervention [29,30,33,35,40,41,43,45,46,47,48,49,50]. No studies empirically evaluated the effect of participant engagement on the results of the study or effectiveness of the intervention. Since all studies that reported outcomes reported positive outcomes, the relationship between stage of end-user consultation and intervention effectiveness was explored. The highest percentage (75%) of studies that showed positive outcomes were studies that involved end-users in prototype testing. This was followed by studies that assessed user needs to inform intervention focus (67%) and those that involved end-users in pilot testing (67%). None of the studies that assessed user needs to inform the technology used for the intervention found positive outcomes. Six (55%) out of 11 studies which involved multiple phases reported positive outcomes [29,32,33,35,44,45].

#### 3.1.11. Sufficiency of Reporting

Sufficiency of reporting scores ranged from 0 (poorest reporting; *n* = 1 study) [43] to 6 (*n* = 3; highest quality reporting) [24,47,50] out of a maximum score of 7. The median score was 4.

## 4. Discussion

This integrative review identified 22 original research studies or protocols for nutrition/diet intervention studies that featured participant engagement in the design or development and were published in the last decade. Within the participatory design methods and processes used, there was no evidence of the explicit use of co-design; however, some studies utilised co-design techniques and 11 studies engaged participants to a collegiate or collaborative level, indicating genuine partnership and meeting PAR requirements. No studies empirically evaluated the specific impact of participant engagement on intervention effectiveness.

To our knowledge, just one published study has reviewed the use of co-design practices within digital health research [14], while the current study is the first to review co-design practices within nutrition research specifically. Similar to Eyles et al., the participatory techniques reviewed in the current study were varied, ranging from conventional methods including focus groups and surveys, to less conventional methods such as photo-voice. Different methods were used specific to various research contexts. For example, the photo-voice method, a visual technique of capturing participants’ concerns which may be sensitive, is pertinent to the needs of the Indigenous population [53]. The interventions included in the current review varied and substantially focused on community-based programs seeking to provide tools and resources to help people, families, or workplaces to adopt healthier dietary behaviours. 

The research designs used were also varied. Qualitative research was common for studies that were at the earlier stages of design (i.e., to inform intervention development), while pre–post and randomised controlled trials were used to evaluate co-designed interventions. Finally, the research populations and samples included a range of end-user stakeholders from specific communities (e.g., ethnic or cultural groups), typically consisting of intervention end-users, although it was not uncommon to include other stakeholders such as health practitioners or other professionals. Promisingly, a substantial number of studies reached a sufficient level of participation whereby power over decision making is shared, suggesting genuine inclusion in the research process [54]. Overall, the body of research demonstrates a heterogenous application of participatory and co-design research techniques that were adapted to the unique needs and characteristics of the health problem or population at hand. Hence, future participatory research could adopt methods suited to similar contexts and evaluate their suitability where necessary. 

### 4.1. Methodological Considerations of the Included Research

A strength of the included studies was that details of the participatory design methods and stakeholders involved in intervention development were reported sufficiently in many key areas. However, it was challenging to determine timeframes of the intervention development process including the total number and time interval between sessions. Although materials used in the design process were named, most were not adequately described. Insufficiently detailed reporting of methodological considerations is a limitation of PAR research previously identified [14,55,56], highlighting the need for more detailed methodological reporting in this field. Notably, studies which involved collegiate participation (the highest form) did not necessarily translate into sufficient reporting. Similarly, consultative participation (the lowest) did not necessarily equate to insufficient reporting.

In this review, only four studies involved a minority population group (African Americans; Bangladeshi migrants) and only two studies involved Indigenous population groups. These findings are contrary to the general acceptance that participatory design is common and best practice in research involving under-served and/or Indigenous populations [57]; however, absence of reporting in the peer-reviewed literature could be a factor and future reviews should include grey literature. Encouragingly, variation in population groups in the current review suggests participatory approaches are applied broadly across populations.

### 4.2. The Effectiveness of Co-Design in Nutrition Research

A secondary objective of this study was to understand the effectiveness of co-design in nutrition/diet-based interventions. However, no studies identified empirically evaluated the effect of participant engagement on these outcomes. Of the 14 studies that assessed intervention outcomes, all but one reported a statistically significant effect in the desired direction, which may have been a result of publication bias towards successful studies [58]. Nonetheless, this review found that a higher percentage of studies reported positive health or health behaviour outcomes if they involved end-users in prototype testing, pilot testing or assessed their needs to inform the intervention’s focus. For example, Adams et al. utilised photo-voice techniques to guide their intervention design and made sure that it was appropriate for the social contexts and met the cultural and practical needs of local Australian Aboriginal people [24]. In future, research may benefit from greater inclusion of end-users at early stages of research design to preliminarily identify the optimal direction. 

There is a dearth of evidence assessing the association between different modes of participation and nutrition intervention effectiveness. Future studies should include controlled trials of treatments that vary levels of participation and co-design or no participation. Future studies should improve reporting, including deficiencies identified in the current review (see below) and to facilitate future best practices. Additional reporting should seek to cost participation in design and estimate return on investment through long-term follow-up. To date, to our knowledge, no study has examined whether co-design is more effective than traditional approaches to intervention development with evaluations tending to be descriptive in nature and not experimental [8,14]. This highlights the need for robust, empirical evaluations that can evaluate these effects. While a randomised controlled trial approach would help to establish the efficacy of a co-designed intervention, it is likely that a RE-AIM approach that considers translational outcomes as well as efficacy outcomes [59] is more appropriate for assessing the effect of co-design. 

A lack of robust evaluations of the impact of co-design have been noted in related fields. Similar to our findings, a recent rapid review found 11 studies reported on the use of co-design in research within acute healthcare settings and that while many studies provided qualitative and descriptive data regarding the perceived value of co-design, robust evaluations were limited [60].

### 4.3. Strengths and Weaknesses of the Review

A strength of this review was that systematic approaches were used across three different scientific databases with studies independently reviewed in duplicate by two co-authors. The integrative review approach also enabled inclusion of both quantitative and qualitative research including studies at different stages of the research process (e.g., protocols, intervention development papers). The addition of Google Scholar to the search also identified a further study that was included. Despite this, by limiting the search to the last 10 years and to articles published in English only, it remains possible that eligible studies were missed. For example, our search strategy did not capture work reflecting a Kaupapa Māori approach to co-design of health or diet interventions [61], potentially because these approaches and cultural groups conceptualise dietary behaviours within a wholistic, whole-of-health model that our search strategy did not detect. Therefore, future reviews should consider cultural and local differences in language and conceptualisation of health to ensure coverage of different groups. The exclusion of grey literature where co-design work may be more commonly published is another limitation of this review. Additionally, although intervention effectiveness was examined, risk of bias could not be assessed due to variation in outcome measures reported in the included studies. However, reporting practices were analysed, which is a valuable outcome of this review.

### 4.4. Implications for Future Research 

This review has several implications for future research. Reporting practices around participatory research have previously been reported to be poor, highlighting the need for researchers to use standard checklists for reporting interventions designed using participatory or co-design methods. Eyles et al. highlighted that checklists can be adapted from existing relevant and appropriate checklists [14], such as those described by Hoffman et al. [62] or Borek et al. [63]. Sufficiency of reporting can provide clearer guidance for future studies to employ methods that are replicable and consistent. Furthermore, it is important to note that co-design techniques and tools are often adapted to the specific research questions, contexts, and populations at hand. Future research would benefit from more open and detailed descriptions of these adaptation processes and the rationales that underpin co-design decision making. To take this even further, it would be ultimately beneficial if researchers begin to openly publish their co-design techniques, similar to how datasets and survey instruments (for example) are increasingly published in open-source libraries, where they can be accessed by researchers and amended for other research purposes.

Other than that, it is suggested that researchers give a higher level of consideration to the time and resources required to design interventions within participatory research. It is important to think about the range of multilevel stakeholders’ representatives that researchers plan to invite to a co-design activity and consider carefully what their drivers and motivations to participate might be. Co-design does not have to be undertaken independently from other research methods; in fact, it works well with other quantitative methods as part of a mixed methods model. Lastly, to determine whether co-design is more effective than traditional approaches to intervention development, high-quality process evaluations and randomised controlled trials should be conducted to assess intervention effectiveness compared to non-co-designed comparator interventions or waitlist control groups.

## 5. Conclusions

Reviews summarising the methods and processes used in participatory and co-design of dietary interventions remain limited. The 22 studies included in this review used participatory research, but not co-design, methods. More studies reported positive health or health behaviour outcomes if they involved participants in prototype testing, pilot testing or needs assessment to inform the intervention focus. Most of the studies did not achieve an adequate level of reporting for their intervention development processes. Further research to explore co-designed nutrition/diet interventions and their effectiveness is warranted.

## Figures and Tables

**Figure 1 nutrients-13-03593-f001:**
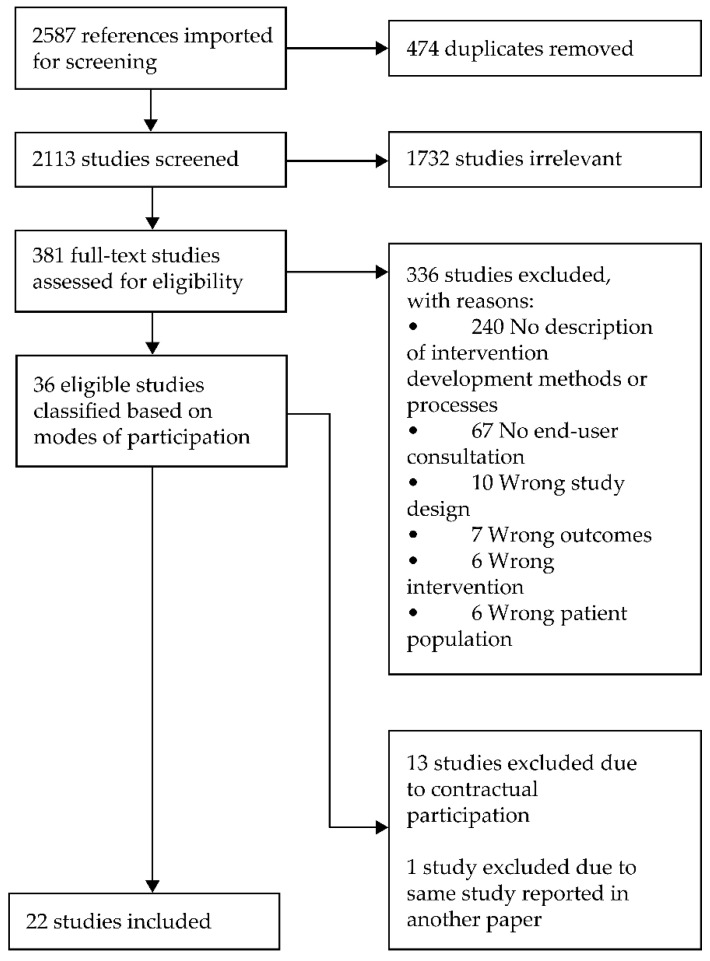
Identification and selection of studies.

**Table 1 nutrients-13-03593-t001:** Definitions, explanations, and eligibility of different levels of participation as described by Cornwall and Jewkes [19] and Biggs [20].

	Definition	Further Explanation	Eligible for Review
Collegiate (deepest form of participation)	Researchers and local people work together as colleagues with different skills to offer, in a process of mutual learning where local people have control over the process.	Deepest level of participation. Researcher’s role shifts from director to facilitator and catalyst.	√
Collaborative	Researchers and local people work together on projects designed, initiated, and managed by researchers.	Collegiate techniques are applied but are influenced by institutional agendas. Genuine participation occurs within the confines of a larger, pre-designed research process.	√
Consultative	People are asked for their opinions and consulted by researchers before interventions are made.	People are involved as informants for the purposes of verifying and amending research findings.	√
Contractual (most shallow form of participation)	People are contracted into the projects of researchers to take part in their enquiries or experiments.	People are involved to fulfil a data collection role and they have no control or input into projects that are scientist-led, designed, and managed.	X

**Table 2 nutrients-13-03593-t002:** Search concepts and synonyms included in searches.

Concept 1: Co-Design	Concept 2: Dietary Intervention
co-design* OR codesign* OR co-creat* OR cocreat* OR “participatory design” OR “design research” OR “collective creativity” OR “user-centred design” OR design* OR “consumer participation” OR pre-design* OR participatory OR “participatory action research” OR “action research” OR “community-based participatory research” OR “co-production” OR “user-centred” OR “human-centred” OR “human-centred design” OR “design thinking” OR “experience based design” OR “experience-based design” OR “experience based co-design” OR “experience-based co-design” OR “experience based codesign” OR “experience-based codesign”	diet* OR nutrition* OR eat OR eating OR food* OR meal* OR “meal plan*” OR menu* adj1 intervention* OR activit* OR strateg* OR program* OR service* OR plan* OR advice OR regime* OR therap* OR provision
AND ⟶	

⟶ The arrow is assumed to be understood as an indicator that synonyms under concept 1 AND synonyms under concept 2 were searched.

**Table 3 nutrients-13-03593-t003:** Study characteristics, co-design methods used, and quality indicators.

Study Reference and Aim	Study Design, Participants and Other Stakeholders, Setting, and Time of Study	Intervention and Main Finding or Outcome	Theoretical Framework and Recruitment Method	Participation Method, Data Collection Techniques, Data Analysis Techniques	Research Stage at Participation Occurred	Quality Indicators
Adams et al. (2012) [24]: To gain an understanding of Aboriginal people’s perspectives on food and food insecurity as an action research method to strengthen food programmes.	Study design: Qualitative (Participatory Action Research). Participants: Men and women (*n* = 10) in their twenties and thirties. Other stakeholders: N/A. Setting: Aboriginal community organisations located in regional Victoria, Australia. Time of study: 2009–2010.	Intervention: Ongoing community-based initiatives to address food security among Aboriginal Australians, including food vouchers. Main outcome: Existing programs adapted to reflect target community’s language, ways of knowing and local challenges or opportunities for healthy eating.	Theory: Core structures of Aboriginal ontology. Recruitment: Invitations through community organisation and health workers; Information session held at community organisation.	Participation method: Photo-voice method. Data collection techniques: Participants took photographs relating to food; focus group discussions; individual interviews for participants storytelling about most significant photographs. Data analysis techniques: Thematic analysis.	Assess background knowledge and evidence, assess user needs to inform intervention focus.	Participatory Action Research standard: Collegiate Standard of Reporting score: 6
Burford et al. (2015) [25]: Utilise participatory design techniques to inform the design of a study that introduces mobile tablet devices in the self-management of type 2 diabetes in a primary healthcare setting.	Study design: Qualitative (Participatory Action Research). Participants: Research team members (*n* = 4); health professionals: general practitioners, specialist, nurses, practice manager (*n* = 11); patients (*n* = 30). Age of participants was not reported. Other stakeholders: N/A. Setting: A general practice super-clinic in Australia. Time of study: Not reported.	Intervention: Mobile tablet devices for the in the self-management of type 2 diabetes in primary healthcare settings. Main outcome: the issue of six “invitations” to 28 people with diabetes to frame their use of a mobile tablet device in managing their health; clustered in themes “Empowered” and “Compelled”, representing typical patient attitudes and behaviours.	Theory: Agency model of customisation for users of new media technologies. Recruitment: Through GP Super Clinic.	Participation method: Facilitated design workshops. Co-design techniques: Examination of available m-health apps and websites, use of iPads to view m-health. Data analysis techniques: Thematic analysis.	Assess user needs to inform intervention focus, assess user needs to inform technology.	Participatory Action Research standard: Collegiate Standard of Reporting score: 3
Sharma et al. (2010) [26]: Describe Health Foods North Programme intervention development and outcomes.	Study design: Qualitative Participants: Inuit and Inuvialuit people (*n* unspecified). Age of participants was not reported. Other stakeholders: Staff from food retailers and local organisations (*n* unspecified). Setting: Community-based; Arctic regions of Nunavut and the NWT, Canada. Time of study: 2008–2009.	Intervention/main outcome: The development of Health Foods North, a culturally appropriate nutrition and physical activity environmental/health promotion intervention addressing chronic disease risk and dietary adequacy.	None reported. Theory: None reported. Recruitment: Posted advertisements and flyers.	Participation method: Interviews and workshops. Data collection techniques: In-depth interviews with community stakeholders, dietary assessment using 24-h recall to target foods for intervention programme, community workshops. Data analysis techniques: Thematic analysis.	Assess user needs to inform intervention focus.	Participatory Action Research standard: Collegiate Standard of Reporting score: 1
Chojenta et al. (2018) [27]: Describe the process of the redevelopment and expansion of Cooking for One or Two, a community-based nutrition education program for older adults.	Study design: Qualitative (focus groups). Participants: Community-dwelling older adults (*n* = 111). Age of participants was not reported. Other stakeholders: Health promotion experts (e.g., a Fellow of the Dietetic Association of Australia); media communication students. Setting: Community-based, large regional city in New South Wales, Australia. Time of study: 2011–2013.	Intervention: Australian-based cooking skills program with education sessions over 5 weeks teaching cooking skills and healthy behaviours while facilitating social interaction. Main outcome: Participants’ experiences informed a supplementary cookbook and education modules. Continued engagement with target group achieved.	Theory: Participatory Action Research (PAR) framework. Recruitment: Past participants and members from a research institute register were recruited.	Participation method: Three-stage iterative intervention development. Data collection techniques: Focus groups and expert consultation, iterative drafting and road-testing of recipe book, telephone interviews, focus group. Data analysis techniques: Not reported.	Develop intervention content, prototype testing, assess user needs to inform intervention focus.	Participatory Action Research standard: Collaborative Standard of Reporting score: 5
Kitzman-Ulrich et al. (2016) [28]: To gather opinions of parent and caregiver dyads on barriers and facilitators, motivators and preferences for a health and weight loss program from a social-ecological perspective.	Study design: Qualitative (Participatory Action Research). Participants: African American parents or caregivers (*n* = 30) with a mean age of 46.1 (SD = 9.8) years, young people (*n* = 25) with a mean age of 12.4 (SD = 1.1) years. Other stakeholders: Graduate students in psychology and public health. Setting: Family-based, South Carolina, USA. Time of study: Not reported.	Intervention: Families Improving Together (FIT): a family- and Social Cognitive Theory- based weight loss intervention. Main outcome: Four main themes established relating to the development of the FIT intervention, e.g., using a positive health promotion framework for weight loss programs, social support.	Theory: Social Cognitive Theory. Recruitment: Through paediatric clinics and community-based organisations.	Participation method: Focus groups. Data collection techniques: Focus groups exploring Social Cognitive Theory predictors of weight loss. Data analysis techniques: Content analysis.	Assess user needs to inform intervention focus.	Participatory Action Research standard: Collaborative Standard of Reporting score: 5
van Dongen et al. (2017) [29]: To adapt an existing experimental nutrition and exercise intervention for frail elderly people to a real-life setting; To test the feasibility and potential impact of this prototype intervention in the new setting.	Study design: Qualitative (Participatory Action Research). Participants: Dietitians and physiotherapists (*n* = 8); Interview participants from the original intervention (*n* = 13) and possible future participants (*n* = 9); Community-dwelling (*n* = 25) elderly ≥65 years (74.1 ± 6.8 years); healthcare professionals including dietitians, physiotherapists, coordinator (*n* = 7); focus group participants (*n* = 14). Other stakeholders: N/A. Setting: Community-based, Harderwijk, The Netherlands. Time of study: Not reported.	Intervention: Nutrition and resistance-type exercise training intervention seeking to improve muscle mass, strength, and physical performance in (pre-)frail older adults. Main outcomes: Successful adaptation of the experimental intervention into real-life settings; intervention was perceived as highly acceptable. Adaptations mostly related to the design of training for implementing and recruiting professionals, design of a dietitian-guided nutrition programme, and organisation of the training sessions.	Theory: Intervention Mapping. Recruitment: Through community nurses from the care organisation and through local organisations and local newspaper ad.	Participation method: 6-stage intervention mapping process followed by pre–post pilot testing of intervention. Data collection techniques: Literature review, semi-structured interviews, focus groups, iterative discussion of findings, pre–post pilot study with interviews and focus groups. Data analysis techniques: Thematic analysis.	Assess user needs to inform intervention focus, prototype testing, pilot/real-world testing.	Participatory Action Research standard: Collaborative Standard of Reporting score: 4
Velema et al. (2018) [30]: To examine effects of a healthy worksite cafeteria (“worksite cafeteria 2.0”] intervention on food purchases. Related works: Velema et al. (2017), Velema et al. (2019) [31,32]	Study design: Randomised Controlled Trial. Participants: Primary outcome was sales data (unspecified *n*) from 30 cafeterias; 1651 employees. Age of participants was not reported. Other stakeholders: Expert interviews (*n* = 14) and seven focus groups (*n* = 45). Setting: Worksite cafeterias in The Netherlands. Time of study: 2016.	Intervention: Workplace cafeteria nutrition intervention seeking healthier purchases in the worksite cafeteria. Main finding: Significant, positive effects of the intervention on purchases for three of the seven studied product groups: healthier sandwiches, healthier cheese as a sandwich filling, and the inclusion of fruit. Increased sales of healthier meal options maintained through 12-week intervention.	Theory: None reported. Recruitment: Participants were recruited by clients (companies) catered by the partnered contract catering companies.	Participation method: Focus groups to inform intervention design. Data collection techniques: Focus groups. Data analysis techniques: Thematic analysis.	Assess background knowledge and evidence, assess user needs to inform intervention focus.	Participatory Action Research standard: Collaborative Standard of Reporting score: 5
Staffileno et al. (2015) [33]: Describe process of adapting a face-to-To adapt a lifestyle change intervention from face-to-face to web-based.	Study design: Mixed methods. Participants: African American adults (18–45 years old) with pre-hypertension. Focus group and survey (*n* = 11); prototype testing (*n* = 8); beta testing (*n* = 8). Other stakeholders: None reported. Setting: Rush University. Medical Center (hospital): USA. Time of study: Not reported.	Intervention: 24 eHealth learning modules that use interactive and situational learning technology. Includes 12 modules focused on Dietary Approaches to Stop Hypertension (DASH) eating plan and 12 on lifestyle physical activity. Main finding/outcome: Successful transformation of face-to-face content into a Web-based platform.	Theory: Social Cognitive Theory; Self-directed behaviour change (behavioural self-management). Motivational coaching. Recruitment: Internet advertisement, print materials, and at blood pressure screenings.	Participation method: Iterative intervention development and pilot testing. Data collection techniques: Focus groups, intervention, development/conversion from face-to-face to eHealth modules, prototype testing in interactive workshop session, pilot testing. Data analysis techniques: Thematic analysis.	Assess user needs to inform intervention focus, prototype testing, pilot/real-world testing.	Participatory Action Research standard: Collaborative Standard of Reporting score: 5
Ard et al. (2010) [34]: Evaluate the effectiveness of a culturally enhanced ‘EatRight’ dietary intervention among African American women in a workplace setting. Related works: Zunker et al. (2008) [35]	Study design: Sequential, control to intervention cross-over design. Participants: Trial participants (*n* = 37) with baseline age of 47.5 (11.8) years Other stakeholders: African American women (*n* = 14) took part in focus groups to inform the research [35]. Setting: Workplace, USA. Time of study: 2006.	Intervention: Culturally modified EatRight Program, based on the concept of “time-calorie displacement”, with large quantities of high-bulk, low-energy-density foods and moderation in high-energy-density foods. Main finding/outcome: Intervention was associated with significant weight loss. Feasibility of the cultural adaptations was established.	Theory: None reported. Recruitment: Pay-check mailers and flyers distributed at headquarters and posted at worksites.	Participation method: Iterative intervention development and pilot testing. Data collection techniques: Nominal Group Technique group discussions, iterative intervention development. Data analysis techniques: Thematic analysis.	Assess background knowledge and evidence, pilot/real-world testing.	Participatory Action Research standard: Collaborative Standard of Reporting score: 5
De Brito-Ashurst et al. (2013) [36]: Describe a theoretical approach to inform the development of a nutrition education programme for adult UK-Bangladeshi chronic kidney disease (CKD) patients. Related works: De Brito-Ashurst et al. (2009), De Brito-Ashurst et al. (2011) [37,38]	Study design: Descriptive Participants: Bengali origin, renal disease patients who participated in a program pilot (*n* = 6). Age of participants was not reported. Other stakeholders: Interpreters, Bengali key workers and local community dietitians; focus group participants (*n* = 20). Setting: East London. UK Time of study: Not reported.	Intervention: 6-month, low-salt dietary behavioural programme consisting of multiple interactions with programme staff and fortnightly telephone calls to reinforce health message. Main outcome: Successful description of the intervention development process.	Theory: Intervention Mapping; PRECEDE model Recruitment: Not reported.	Participation method: Intervention mapping and PRECEDE approach. Data collection techniques: Literature review, focus groups Co-design data analysis techniques: Not reported.	Assess background knowledge and evidence, assess user needs to inform intervention focus.	Participatory Action Research standard: Collaborative Standard of Reporting score: 3
Franco et al. (2013) [39]: To conduct impact evaluation of activities to promote fruit and vegetables (FV) consumption in the workplace.	Study design: Before-after. Participants: Workers who had lunch in the workplace cafeteria during the study, *n* = 197 (mean age = 40 (8.3) years). Other stakeholders: Concessionaire owner and nutritionist. Setting: workplace in Rio de Janeiro, Brazil. Time of study: 2007–2009	Intervention: 9-month program involving environmental and educational components (e.g., provision of educational material, food tasting stand). Main finding: On average, the coverage of educational activities and materials was 63.5%. FV consumption increased by 38% in employees.	Theory: Not reported. Recruitment: None reported.	Participation method: Focus groups to inform intervention design. Data collection techniques: Focus groups, intervention development considering stakeholder preferences/needs. Data analysis techniques: Not specified.	Assess background knowledge and evidence, assess user needs to inform intervention focus, pilot/real-world testing.	Participatory Action Research standard: Collaborative Standard of Reporting score: 2
Hemmingsson et al. (2012) [40]: To evaluate weight loss and the dropout rate after a 1-year commercial weight loss program.	Study design: Observational cohort study. Participants: Enrolled customers in a weight loss program with a mean age of 48 ± 12 years (range: 18–81 years) Other stakeholders: None reported. Setting: Sweden. Time of study: Not reported.	Intervention: 1-year structured weight loss support program with 1-h group sessions. Very low calorie diet, low calorie diet, or restricted normal-food diets offered. Main finding: After 1 y, mean (±SD) weight changes were −11.4 ± 9.1 kg with the VLCD (18% dropout), −6.8 ± 6.4 kg with the LCD, and −5.1 ± 5.9 kg with the restricted normal-food diet.	Theory: None reported. Recruitment: Not specified.	Participation method: Tailoring of intervention to participants’ health goals, food preferences, and nutritional requirements. Data collection techniques: Interview/discussion between participant and health coaches. Decision was based on baseline BMI, desired weight loss, and personal preference.Data analysis techniques: Not specified.	Assess user needs to inform intervention focus	Participatory Action Research standard: Consultative Standard of Reporting score: 3
Hernandez et al. (2014) [41]: To evaluate the effects of a diet high in total carbohydrate (higher-complex, lower glycaemic index [GI]) and minimal fat on control of maternal glycemia and postprandial lipids. Related works: Hernandez et al. (2016) [42]	Study design: Quantitative (Randomised crossover trial) Participants: Women with diet-controlled gestational diabetes mellitus (GDM), *n* = 16, 28.4 ± 1.0 years. Other stakeholders: None reported. Setting: University Hospital, Kaiser Permanente Colorado Institute; Colorado, USA. Time of study: Not reported.	Intervention: Higher-complex carbohydrate (HCC) and lower-fat (LF) ‘Choosing Healthy Options In Carbohydrate Energy’ (CHOICE) diet. Main finding: A diet high in complex carbohydrates and limited fat was effective in controlling maternal glycemia to within current recommended ranges.	Theory: Not reported Recruitment: Not reported	Participation method: Tailoring of intervention to participants’ health goals, food preferences, and nutritional requirements. Data collection techniques: Food frequency questionnaire completed to establish calorie requirements for individual participants. Data analysis techniques: Descriptive.	Assess user needs to inform intervention focus	Participatory Action Research standard: Consultative Standard of Reporting score: 2
Hiel et al. (2019) [43]: To evaluate the impact of daily consumption of inulin-rich vegetables on gut microbiota, gastrointestinal symptoms, and food-related behaviour in healthy individuals.	Study design: Quantitative—single group-design trial Participants: Healthy adults (*n* = 25) aged 21.84 ± 0.39 years Other stakeholders: None reported. Setting: Université Catholique de Louvai, Belgium. Time of study: Not reported.	Intervention: Dietary intervention including inulin-type fructans (ITFs)-rich vegetables to reach a minimum intake of at least 9 g ITF/d in healthy volunteers Main finding: Higher consumption of ITF-rich vegetables associated with increase in well-tolerated dietary fibre.	Theory: None reported. Recruitment: Not reported.	Participation method: Tailoring of intervention to participants’ previous intake/acceptability of vegetables. Data collection techniques: Food diaries, fasting breath samples, visual analogue scales, stool samples. Data analysis techniques: Not specified.	Assess user needs to inform intervention focus	Participatory Action Research standard: Consultative Standard of Reporting score: 0
Jacobsson et al. (2012) [44]: To examine the impact of active patient education on gastrointestinal symptoms in women with a gluten-free diet.	Study design: Quantitative (Randomised controlled trial) Participants: Women with coeliac disease (*n* = 106), mean age = 53 years, range = 23–80 years Other stakeholders: PBL expert supervised instructors. Setting: Hospitals, Southeast Sweden. Time of study: Not reported.	Intervention: 10-session educational program to support and encourage self-identification of lifestyle changes to reduce gastrointestinal (GI) symptoms and explore new knowledge. Main finding: “Celiac School” participation was associated with significant improvements.	Theory: Problem-based learning Recruitment: Not reported	Participation method: Problem-based learning. Data collection techniques: Weekly meetings in groups of 7–9 persons conducted by a tutor familiar with PBL, self-report questionnaires. Data analysis techniques: Not specified.	Assess user needs to inform intervention focus, develop intervention content.	Participatory Action Research standard: Consultative Standard of Reporting score: 4
Kim et al. (2013) [45]: To translate and validate a culturally modified DASH for Koreans (K-DASH) and gather preliminary evidence of efficacy.	Study design: Mixed methods with pre–post intervention evaluation design. Participants: Korean Americans (*n* = 30), mean age = 55.3 (6.8) years Other stakeholders: Clinicians; community health workers were involved in group education sessions. Setting: Centrally located community-based organisation, The Korean Resource Centre, in the Baltimore-Washington metropolitan area, USA. Time of study: 2011	Intervention: 10-week culturally modified K-DASH intervention consisting of two structured in-class education sessions with interactive group activities, 3 individually tailored nutrition consultations with a bilingual nurse/dietician team, and 1 follow-up telephone call. Main finding: Both systolic blood pressure and diastolic were significantly decreased at postintervention evaluation. A culturally relevant and efficacious dietary intervention was produced.	Theory: Community-based participatory research Recruitment: Advertisements, personal networks, referrals from community physician networks.	Participation method: Community-based participatory action research. Data collection techniques: Needs analysis, review of evidence, focus groups, pre–post intervention evaluation. Data analysis techniques: Not specified.	Assess user needs to inform intervention focus, pilot/real-world testing.	Participatory Action Research standard: Consultative Standard of Reporting score: 3
Madjd et al. (2016) [46]: To compare the effect of high energy intake at lunch with that at dinner on weight loss and cardiometabolic risk factors in women during a weight loss program.	Study design: Quantitative (Randomised clinical trial) Participants: Overweight or obese women (*n* = 80), 18–45 years. Other stakeholders: None reported Setting: NovinDiet weight loss clinic, Iran. Time of study: Not reported.	Intervention: Hypoenergetic diet: high-carbohydrate, low-saturated fat diet, with ≥400 g fruit and vegetables to achieve fibre intake recommendation of 25 g/day]. Main meal consumed either at lunch (LM) or dinner (DM). Main finding/outcome: Compared with the DM group, LMgroup had greater reductions in weight and BMI.	Theory: Stages of change model Recruitment: Not specified.	Participation method: Tailoring of intervention to participants’ food diaries and preferences. Data collection techniques: Anthropometric measurements, blood samplesData analysis techniques: Not specified.	Assess user needs to inform intervention focus	Participatory Action Research standard: Consultative Standard of Reporting score: 2
Mosher et al. (2013) [47]: To examine whether changes in self-efficacy explain the effects of a mailed print intervention on long-term dietary habits among breast and prostate cancer survivors.	Study design: Quantitative (Randomised trial) Participants: Diagnosed with early-stage breast or prostate cancer within the prior nine months (*n* = 543), mean age = 57.2 (10.7) years Other stakeholders: None reported. Setting: Community-based; North America. Time of study: Not reported.	Intervention: FRESH START 10-month mailed print interventions focused to improve diet and PA; based on Social Cognitive Theory. Main finding: Change in self-efficacy for fat restriction partially explained effect on fat intake; change in self-efficacy for F&V consumption partially explained change in daily F&V intake.	Theory: Social Cognitive Theory; Observational learning Recruitment: Cancer registries of participating medical centres, large oncology practices, or self-referral.	Participation method: Tailoring of intervention content to participants’ current diet and physical activity behaviours and other factors. Data collection techniques: Diet History Questionnaire, 7-day Physical activity Recall, self-efficacy. Data analysis techniques: Descriptive.	Assess user needs to inform intervention focus	Participatory Action Research standard: Consultative Standard of Reporting score: 6
Nybacka et al. (2017) [48]: To examine the effects of diet and exercise interventions on metabolic profile and cardiovascular risk factors women with polycystic ovary syndrome (PCOS).	Study design: Quantitative (Randomised controlled trial) Participants: Women with PCOS (*n* = 57) 18–40 years Other stakeholders: None reported. Setting: Women’s Health Research Unit, Karolinska University Hospital, Stockholm, Sweden. Time of study: Not reported.	Intervention: Diet, exercise, or diet + exercise 16 week program. Diets individually designed with dietitian, seeking 600 kcal/day reduction in calorie consumption. Main finding/outcome: BMI, waist circumference and total cholesterol significantly reduced in diet and diet + exercise groups.	Theory: None reported. Recruitment: Not reported.	Participation method: Tailoring of intervention to suit participants’ individual nutritional requirements and food preferences. Data collection techniques: Self-reported food intake, pedometer, fasting blood test, DEXA scan. Data analysis techniques: Not specified.	Assess user needs to inform intervention focus	Participatory Action Research standard: Consultative Standard of Reporting score: 2
Rudel et al. (2011) [49]: To evaluate the contribution of food packaging to exposure to Bisphenol A (BPA) and bis(2-ethylhexyl) phthalate (DEHP) chemicals used in food packaging.	Study design: Quantitative (Quasi-experimental pre–post design) Participants: Family members with exposure to BPAs (e.g., consumed canned foods): *n* = 20. Median age of the 10 adults was 40.5 years, median age of the 10 children was 7 years. Other stakeholders: Caterer. Setting: Community-based, San Francisco Bay Area, USA. Time of study: Not reported.	Intervention: A three-day special diet of fresh foods (no canned foods) prepared and packaged almost exclusively without contact with plastic. Main finding/outcome: The fresh foods intervention reduced geometric mean concentrations of BPA by 66% and DEHP metabolites by 53–56%.	Theory: Not reported Recruitment: Letters sent via listservs.	Participation method: Stakeholder input into menu design. Data collection techniques: Urine samples, daily phone calls with research staff, food questionnaires. Data analysis techniques: Not specified.	Assess user needs to inform intervention focus	Participatory Action Research standard: Consultative Standard of Reporting score: 3
Shahar et al. (2012) [50]: To develop nutrition education materials to promote healthy aging and reducing risk of chronic diseases in older adults living in a rural area.	Study design: Qualitative (Participatory Action Research) Participants: Older adults (≥60 years old; *n* = 33); Health professionals, e.g., rural clinic staff, physicians, medical assistants, nurses (*n* = 14) with a mean age of 30.9 ± 8.3 years. Other stakeholders: A professional artist; dietitians, nutritionists, public health physicians and anthropologist. Setting: Health clinics in Klang Valley, Malaysia. Time of study: Not reported.	Intervention: A nutrition education package (booklet, flipchart, and placemats). Main finding/outcome: A total of 42.4% of the older adults expressed that the sentences in the flipchart needed to be simplified and medical terms explained. Terminology, illustrations, and nutrition recommendations were barriers to understanding of educational materials.	Theory: None reported. Recruitment: Not specified	Participation method: Three stage-approach: Needs assessment, intervention development, evaluation (prototype testing). Data collection techniques: self-administered questionnaire. Data analysis techniques: Descriptive analysis.	Prototype testing	Participatory Action Research standard: Consultative Standard of Reporting score: 6
Uddin et al. (2017) [51]: To develop and test a mobile phone-based system to improve health, population and nutrition services in rural Bangladesh and evaluate its impact on service delivery.	Study design: Quantitative (Quasi-experimental pre–post design). Participants: Target population: currently married women of reproductive age. Other stakeholders: Service-delivery personnel, health, and planning officers. Setting: two administrative divisions of Bangladesh. Time of study: Not applicable.	Intervention: Mobile phone-basedsystem to improve health, population, and nutrition services in rural Bangladesh. Main finding/outcome: Establishment of a research protocol.	Theory: None reported. Recruitment: Routine community visits of health and family planning workers.	Participation method: Intervention designed with input (feedback) from stakeholders. Data collection techniques: Surveys. Data analysis techniques: Not specified.	Assess user needs to inform intervention focus, pilot/real-world testing.	Participatory Action Research standard: Consultative Standard of Reporting score: 4

## Data Availability

Not applicable.

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
