# Peer review of "Co-Design Practices in Diet and Nutrition Research: An Integrative Review"

_nutrients, 2021, doi:10.3390/nu13103593_

Round 1
Reviewer 1 Report
This is an excellent review on co-design practies in diet and nutrition research including integrative review methodology. Very timely.
My main comment is that there is no reporting on co-design studies using kaupapa Māori approach with indigenous Māori in Aotearoa New Zealand. Maybe it didn't quite fit the criteria? I think this would have really added value to this review.
Reviewer 2 Report
This manuscript under consideration is well written and covers an important topic. Below please find a few suggestions to help strengthen the manuscript.
1) l14 Abstract Please briefly define "co-design"
2) Abstract please add number of studies identified in the initial search
3) l23 Please rephrase the sentences starting with numbers and use active rather than passive voice e.g. "We included ..."
4) l170 Methods Would it be possible to briefly perform an updated search since the search was conducted over a year ago in the meantime
5) Figure 1 please reformat so that "22 studies included" is at the bottom. Did you handsearch? This additional method often yields 10% of included studies as formal searches may miss key articles.
6) Table 3 should include year the study was conducted. Currently the order is not obvious. This type of table may be organized by first authors last name, year or publication or thematically. Consider including age of participants.
7) L281 This line refers to 23 studies while the rest of the manuscript and figures refer to 22.
8) Table 4 consider merging with Table 3 and changing the layout to horizontal. Currently it is difficult to interpret the information in Table 4 without basic study information included in Table 3.
9) Would it be possible to include a visual or overview summary figure or table
10) Please add the year the study was published
Round 2
Reviewer 2 Report
The authors have revised the manuscript
sufficiently addressing all comments previously
raised.